# Anti-CD20 Antibody and Calcineurin Inhibitor Combination Therapy Effectively Suppresses Antibody-Mediated Rejection in Murine Orthotopic Lung Transplantation

**DOI:** 10.3390/life13102042

**Published:** 2023-10-11

**Authors:** Hiroki Matsumoto, Hidemi Suzuki, Takahiro Yamanaka, Taisuke Kaiho, Atsushi Hata, Terunaga Inage, Takamasa Ito, Toshiko Kamata, Kazuhisa Tanaka, Yuichi Sakairi, Shinichiro Motohashi, Ichiro Yoshino

**Affiliations:** 1Department of General Thoracic Surgery, Graduate School of Medicine, Chiba University, Chiba 260-8670, Japan; asdf7587@yahoo.co.jp (H.M.); ty_ranappu_speed@yahoo.co.jp (T.Y.); kihutisk826@yahoo.co.jp (T.K.); potatolunch@yahoo.co.jp (T.I.); kazutanaka1118@yahoo.co.jp (K.T.); y_sakairi1@yahoo.co.jp (Y.S.); iyoshino@faculty.chiba-u.jp (I.Y.); 2Department of Thoracic Surgery, Kimitsu Chuo Hospital, 1010 Sakurai, Kisarazu 292-8535, Japan; 3Department of General Thoracic Surgery, Chiba Cancer Center, Chiba 260-8717, Japan; atsushata@gmail.com (A.H.); takamasaito10211013@gmail.com (T.I.); 4Department of Thoracic Surgery, International University of Health and Welfare Atami Hospital, Shizuoka 413-0012, Japan; tkamata-cib@umin.ac.jp; 5Department of Medical Immunology, Graduate School of Medicine, Chiba University, Chiba 260-8670, Japan; motohashi@faculty.chiba-u.jp; 6Department of General Thoracic Surgery, International University of Health and Welfare Narita Hospital, Chiba 286-8520, Japan

**Keywords:** anti-CD20 antibodies, calcineurin inhibitors, antibody-mediated rejection, lung transplantation, donor specific antibodies

## Abstract

Antibody-mediated rejection (AMR) is a risk factor for chronic lung allograft dysfunction, which impedes long-term survival after lung transplantation. There are no reports evaluating the efficacy of the single use of anti-CD20 antibodies (aCD20s) in addition to calcineurin inhibitors in preventing AMR. Thus, this study aimed to evaluate the efficacy of aCD20 treatment in a murine orthotopic lung transplantation model. Murine left lung transplantation was performed using a major alloantigen strain mismatch model (BALBc (H-2d) → C57BL/6 (BL/6) (H-2b)). There were four groups: isograft (BL/6→BL/6) (Iso control), no-medication (Allo control), cyclosporine A (CyA) treated, and CyA plus murine aCD20 (CyA+aCD20) treated groups. Severe neutrophil capillaritis, arteritis, and positive lung C4d staining were observed in the allograft model and CyA-only-treated groups. These findings were significantly improved in the CyA+aCD20 group compared with those in the Allo control and CyA groups. The B cell population in the spleen, lymph node, and graft lung as well as the levels of serum donor-specific IgM and interferon γ were significantly lower in the CyA+aCD20 group than in the CyA group. Calcineurin inhibitor-mediated immunosuppression combined with aCD20 therapy effectively suppressed AMR in lung transplantation by reducing donor-specific antibodies and complement activation.

## 1. Introduction

Lung transplantation is a curative treatment for patients with various end-stage lung diseases. However, survival after lung transplantation remains shorter than that after other solid organ transplantations [1]. Adult recipients who undergo lung transplantation have a median survival of 5.6 years, with unadjusted survival rates of 53% at 5 years and 31% at 10 years [2].

The leading cause of death after lung transplantation is chronic lung allograft dysfunction (CLAD), which remains poorly understood. There have been no effective preventive measures or therapeutic interventions targeting CLAD [3,4]. Antibody-mediated rejection (AMR) is one of the most critical risk factors for CLAD and consequent poor prognosis [5,6,7,8,9].

AMR following lung transplantation was only recently defined in 2016 [10]. Immunosuppressive therapy, mediated by calcineurin inhibitors (CNIs), is essential after lung transplantation but insufficient to prevent or treat AMR. The central concept of AMR involves immune activation, wherein allospecific B cells and plasma cells produce antibodies against donor lung antigens. Progress has been made in recent years concerning AMR after lung transplantation, but it still lags behind research on other organ transplantations [11,12].

Consequently, treatment options for AMR are limited, and the efficacy of each drug has been poorly reported. Using rodent models of lung transplantation, our laboratory has previously reported antigen–antibody reactions, such as anti-collagen type V antibody [13], minor antigen [14,15], and complement activation [16,17].

Rituximab is an anti-CD20 antibody (aCD20) used in combination therapy to prevent and treat AMR. There is widespread clinical use of rituximab for desensitization in both ABO blood group- and HLA antibody-incompatible patients prior to renal transplantation [18,19]. The introduction of rituximab has significantly improved the outcomes of human ABO-incompatible liver and renal transplantations [20,21]. Some reports have documented rituximab use in lung transplantation [6,9,22,23,24]. However, to the best of our knowledge, no reports or prospective clinical trials have evaluated the efficacy of single additional use of aCD20 in preventing AMR. This study examined the efficacy of adding aCD20s to CNI in a murine lung transplantation model.

This study aimed to evaluate the efficacy of aCD20 treatment in preventing AMR in a murine orthotopic lung transplantation model.

## 2. Materials and Methods

### 2.1. Animal Model

BALB/c(H2d) and C57BL/6(H2b) mice (CLEA Japan, Inc., Tokyo, Japan) were used for orthotopic left lung transplantation. All experiments utilized male mice. Under institutional guidelines, the mice were housed at the Biomedical Research Center of the Graduate School of Medicine, Chiba University. The BALB/c(H2d) and C57BL/6(H2b) mice were utilized as donors, aged between 6 and 12 weeks (body weight, 24–32 g), and C57BL/6(H2b) mice were utilized as recipients, aged between 7–12 weeks (body weight, 24–32 g). This study was approved by the Institute for Animal Care at Chiba University (approval code: A3-010).

### 2.2. Surgical Technique

Orthotopic left lung transplantation was performed using a previously described sterile technique [14]. Briefly, a left lung graft was harvested from a donor mouse. Cuffs made of 20–24-gauge intravenous catheters were inserted into the pulmonary artery, pulmonary vein, and bronchus of the grafted left lung. The graft was fixed and then implanted in place of the left lung of the recipient mouse. Buprenorphine (0.1 mg/kg) and saline solution (1 mL) were injected subcutaneously for pain management and rehydration during general anesthesia.

There were four groups: three major alloantigen strain mismatch models and one isogenic transplant model (Iso control), in which C57/BL6 mice were used as donors and recipients. The allogeneic lung transplantation models were divided into three groups: no medication (Allo control), cyclosporine A (CyA; Tokyo Chemical Industry Co., Tokyo, Japan) (CyA group) treatment, and CyA and murine aCD20 (clone SA271G2 Rat IgG2bκ, BioLegend, San Diego, CA, USA) (CyA+aCD20 group) treatment groups.

The mice were sacrificed 14 days after lung transplantation. General anesthesia was induced in the recipient mice using Isoflurane. Blood samples were collected from the inferior vena cava, and the spleen was removed at the time of sacrifice. The lungs and mediastinal lymph nodes were removed through a median sternotomy after heparin-filled saline was injected into the pulmonary artery to flush out the blood. The serum was separated by centrifugation and stored at −80 °C until use. Each group comprised at least five (five to nine) animals.

### 2.3. Medication Protocols

The CyA and CyA+aCD20 groups were treated daily with 10 mg/kg CyA by hypodermic injection, starting on the day of transplantation. In the CyA+aCD20 group, for B-cell depletion, 250 μg of aCD20 was injected into the tail vein seven days before lung transplantation (Figure 1). The B-cell suppression effect has been previously described [25].

### 2.4. Histology and Immunohistochemistry

The left lung grafts were fixed in 10% formalin. The tissues were embedded in paraffin and sectioned at 4 μm. The paraffin sections were stained with hematoxylin–eosin (HE). HE staining was used to determine the score (Grade-A score [A score] and Grade-B score [B score]) on the basis of the criteria of the International Society for Heart and Lung Transplantation [26,27]. In brief, acute rejection based on perivascular and interstitial mononuclear infiltrates was graded as Grade A0 (none), A1 (minimal), A2 (mild), A3 (moderate), and A4 (severe). Lymphocytic bronchiolitis was graded as Grade B0 (none), B1R (low grade), and B2R (high grade). Paraffin sections were also used for immunohistochemical staining with a rabbit polyclonal antibody to mouse C4d (HP8033, Hycult Biotech, Uden, Netherlands). Capillary C4d staining was scored as previously reported [26,28]. In brief, capillary C4d staining was graded as 0, 1 (<10% of capillaries), 2 (10–50%), or 3 (>50%). The slides were evaluated by three surgeons (HM, HS, and TK) who were blinded to the control or experimental groups to which each slide belonged.

### 2.5. Flow Cytometry

Splenocytes and lymphocytes were obtained by squeezing the spleen and the lymph nodes. Ammonium-Chloride-Potassium lysing buffer was used to remove red blood cells. Single cells on the grafted lung tissue were obtained using a gentleMACS™ Octo Dissociator and the Lung Dissociation Mouse Kit (Miltenyi Biotec, Bergisch Gladbach, Germany). The percentage of CD19-positive cells among lymphocytes defined the B-cell population. Single cells were stained with the following fluorochrome-labeled anti-mouse antibodies: anti-CD3-APC/Cy7 (clone 17A2, BioLegend, San Diego, CA, USA) and anti-CD19-PE (clone 6D5, BioLegend, San Diego, CA, USA). Cells were acquired on the BD LSRFortessa II analyzer (BD Biosciences, San Jose, CA, USA). The mean fluorescence intensity (MFI) was assessed using FlowJo software (BD Life Sciences, Franklin Lakes, NJ, USA).

### 2.6. Measurement of Serum Donor-Specific Antibodies

Donor-specific antibodies (DSAs) in recipient blood were detected by flow cytometric cross match as previously described [29]. Briefly, splenocytes obtained from the donor’s spleen were prepared and incubated in recipient serum samples for 30 min. After washing, the cells were incubated with goat anti-mouse IgG (ab6785; Abcam, Cambridge, UK) and IgM (ab97229; Abcam, Cambridge, UK), along with anti-mouse CD3-APC/Cy7 antibodies (clone 17A2, BioLegend, San Diego, CA, USA), for T-cell gating by flow cytometry. Subsequently, the cells and the antibodies were analyzed using the BD LSRFortessa II analyzer (BD Biosciences, San Jose, CA, USA). MFI was assessed using the FlowJo software.

### 2.7. Cytokine Analysis

Sera from individual recipient mice in the Iso control (n = 5), Allo control (n = 5), CyA (n = 5), and CyA+aCD20 (n = 5) groups were analyzed using the BD Cytometric Beads Array (CBA) System (BD Bioscience, San Jose, CA, USA), as previously described [30]. Serum cytokine levels were measured using the CBA Mouse Th1/Th2/Th17 Cytokine Kit (Becton Dickinson, Franklin Lakes, NJ, USA), CBA Mouse IL-12/IL-23p40 Flex Set (BD Biosciences, San Jose, CA, USA), and CBA Mouse IL-21 Flex Set (BD Bioscience, San Jose, CA, USA). MFI was assessed using the FCAP Array™ v3.0 software (BD Bioscience, San Jose, CA, USA). CBA samples were analyzed in triplicate to provide sufficient data for statistical validation of the results. Values that deviated by more than 20% from the average of the three values were excluded, and the average was recalculated.

### 2.8. Statistical Analysis

All data are presented as mean ± standard error of the mean. When comparing four groups, data were evaluated using the Kruskal-Wallis test and post-hoc Steel-Dwass test. Statistical significance was set at *p* < 0.05. Statistical analyses were performed using EZR (Saitama Medical Center, Jichi Medical University, Saitama, Japan), a graphical user interface for the R statistical program (The R Foundation for Statistical Computing, Vienna, Austria).

## 3. Results

Figure 2 shows representative explanted macroscopic lung grafts and microphotographs at the time of sacrifice. The arrowheads indicate the left graft lung. In the Iso control group, the color tone and dilation of the graft lung were similar to those in the native lung, and there was no infiltration or C4d deposition. In the macrographic comparison, the Allo control group showed color changes due to lung congestion, the CyA group showed slight improvement, and the CyA+aCD20 group showed significant improvement. The lung grafts showed perivascular cellular infiltration in the Allo control and CyA groups, whereas inflammation was improved in the CyA+aCD20 group. C4d deposition was observed in the Allo control and CyA groups, but not in the CyA+aCD20 group.

Figure 3 shows the histological grading: A, B, and C4d scores. There was no significant difference in A and C4d scores between the Allo control and CyA groups (A score, 2.45 ± 0.50 [Allo control] vs. 2.40 ± 0.49 [CyA], *p* = 0.99; C4d score, 2.67 ± 0.47 [Allo control] vs. 2.88 ± 0.33 [CyA], *p* = 0.76). However, these scores were significantly improved in the CyA+aCD20 group compared with the Allo control group (A score, 1.6 ± 0.50 [CyA+aCD20] vs. 2.5 ± 0.50 [Allo control], *p* = 0.019; C4d score, 0.29 ± 0.33 [CyA+aCD20] vs. 2.7 ± 0.47 [Allo control], *p* = 0.0040) and CyA group (A score, 1.55 ± 0.50 [CyA+aCD20] vs. 2.40 ± 0.49 [CyA], *p* = 0.030; C4d score, 0.29 ± 0.70 [CyA+aCD20] vs. 2.88 ± 0.33 [CyA], *p* = 0.0028). The Allo control group exhibited the worst B score. B scores were improved in the CyA+aCD20 group compared with those in the Allo control group (0.22 ± 0.63 [CyA+aCD20] vs. 0.91 ± 0.67 [Allo control], *p* = 0.022), and CyA group (0.22 ± 0.63 [CyA+aCD20] vs. 0.40 ± 0.49 [CyA], *p* = 0.27), but there was no significant difference. These scores and *p*-values for each groups are summarized in Table 1.

The percentage of B cells among lymphocytes decreased in both the spleen and peripheral blood after the administration of aCD20, and this reduction in B cell percentage was sustained for at least 2 weeks. The B cell populations were 0.96% and 23% in the spleen at 3 weeks after aCD20 administration, a significant difference between the two cases. (Figure 4A). The percentage of B cells in various organs was examined 14 days after transplantation. The CyA+aCD20 group showed a significantly lower percentage of B cells than the CyA group. (Figure 4B).

We measured circulating DSA levels in each group using flow cytometry (Figure 5). The levels of each type (IgM and IgG) of DSA in the Allo control group were higher than those in the Iso control group. However, the levels of the DSAs were comparable between the Allo control and CyA groups. The CyA+aCD20 group exhibited significantly lower MFIs for DSA-IgM compared with those in the CyA group. However, the MFI for DSA-IgG was comparable (IgM, 3071.2 ± 666.1 vs. 4176.8 ± 613.2, *p* = 0.0079; IgG, 1605 ± 385.0 vs. 11283.8 ± 4534.1, *p* = 0.095).

Nine serum cytokines were simultaneously measured in triplicates. None of the measurements deviated by >20% from the mean. Among the measured cytokines, only interferon (IFN)-γ levels were significantly lower in the CyA+aCD20 group compared with those in the CyA group (168.75 ± 11.14 pg/mL vs. 198.84 ± 27.20 pg/mL, *p* = 0.0047) (Figure 6).

## 4. Discussion

AMR after lung transplantation occurs in approximately 25% of cases [31] and can lead to acute and chronic graft failure [16,32]. Prophylactic and therapeutic strategies for AMR after lung transplantation have been used in clinical practice to mimic those after kidney and liver transplantation. However, their efficacy has not been sufficiently evaluated. Specifically, combination therapy with methylprednisolone, intravenous immunoglobulin, plasma exchange, and/or rituximab may be administered for AMR to suppress B cells and deplete circulating antibodies [7,10,23]. Multiple drugs are often used simultaneously, and there are few reports on the effectiveness of each treatment.

In the present study, murine aCD20s suppressed AMR after lung transplantation. The Allo control and CyA groups met the diagnostic criteria for clinical AMR, including pathological rejection of the transplanted lung, positive immunostaining for C4d deposition, and high serum DSA levels. The A and C4d scores in the CyA+aCD20 group were significantly suppressed compared with those in the CyA group.

According to previous reports, prophylactic administration of aCD20, in addition to T-cell suppression, is effective in preventing rejection in skin and heart transplantation animal models [33,34]. We tried a similar experiment in the lung transplant model. To evaluate the impact of aCD20 on AMR, CyA was used for T-cell suppression, as in previous reports. Compared with the Allo group, the CyA group did not show significant improvement. However, this may be due to the greater impact of AMR rather than acute cellular rejection or potentially to an insufficient dosage of CyA monotherapy in the present model.

Due to the fact that B cells have multiple functions, including immunoglobulin production, cytokine production, and antigen presentation, therapeutic approaches targeting B-cells are important for controlling AMR [35,36]. Smirnova et al. identified the infiltration of grafts by activated B cells in a mouse model of chronic rejection, showing that B cells are deeply involved in chronic rejection [37]. In an experiment using aCD20, Watanabe et al. used a model of chronic rejection characterized by impaired tissue perfusion. They successfully suppressed the effects of lymphocyte infiltration into the graft using aCD20 [38]. The suppression of B cells not only decreases antibody production but also suppresses T cell activation; these two effects are thought to act synergistically to suppress organ rejection [36,39,40]. The above hypothesis is supported by the fact that both the extent of T cell infiltration and IFN-γ levels produced by T cells were suppressed in this experiment. Figure 4A shows that B cells were completely suppressed at the time of lung transplantation. The timing of the sacrifice was three weeks after aCD20 administration. In Figure 4A, on day 21, the B-cell population in the spleen was quite different between the two cases (0.96% and 23%). According to the original paper [25], the effect of the anti-CD20 antibody lasted a little longer, so it is possible that the dosage in this experiment was reduced slightly due to the residual drug in the syringe or needle, or the effect differed depending on the individual mouse. As depicted in Figure 4B, the B cell suppression in each organ was mild compared to that in Figure 4A. However, we thought that aCD20 had an effect on B cell suppression in all organs, and the effect weakened a little with time since we observed similar reductions in all organs. The reason why the CyA+aCD20 group in Figure 4B showed less B cell suppression compared to Figure 4A could be due to errors in drug administration or the effects of surgical invasion. In addition, Figure 4B shows that the error bars are larger in three major alloantigen strain mismatch transplant models compared to the Iso group. In Figure 4B, some of the error bars were large regardless of drug administration, suggesting that major alloantigen strain mismatch transplant invasion is likely to affect the percentage of B cells beyond the effect of B cell suppression by aCD20. Continuous administration of the aCD20 for B cell suppression up to the time of sacrifice might have further prevented the rejection.

When the CyA+aCD20 group was compared with the CyA group, DSA-IgM, but not DSA-IgG, levels were significantly lower. Generally, the time of peak serum IgM and IgG levels from the initiation of murine humoral immune reaction is different: 14 days for IgM and 28–49 days for IgG [41]. This explains the absence of a difference in DSA-IgG levels between the groups in the present model. Furthermore, the increase in DSA-IgG levels over time could not be observed in this experiment. According to Sweet et al., perioperative administration of rituximab reduced de novo DSA in a randomized, placebo-controlled, double-blind study [42]. DSA reduction is expected to be effective in reducing cases of CLAD. This mouse model was similarly effective in reducing DSA, and further studies with rituximab are expected in the future.

Stroopinsky et al. reported that patients with non-Hodgkin lymphoma receiving rituximab showed a significant decline in IL-2 and IFN-γ levels and inflammatory T cells in peripheral blood [43]. In the present study, IFN-γ levels declined significantly, and IL-2 levels trended downward but not significantly in the CyA+aCD20 group compared with the CyA group. The complex relationship between B and T cells might explain why other T cell-produced cytokines were not reduced. Marino et al. reported that anti-CD20-mediated B cell depletion prevented the generation of CD4 memory T cells while enhancing that of CD8 memory T cells after transplantation in mice. These experiments suggest that the effect of aCD20 treatment is mainly attributable to the reduction in DSA levels, not cytokine modulation. There are few reports on the association between cytokines and AMR, and further studies are needed.

In the present study, C4d deposition was observed in all recipients in the Allo control and CyA groups, and aCD20 treatment markedly improved the C4d score. However, a consensus report on AMR concluded that C4d deposition is not necessary for diagnosing AMR in lung transplantation [10]. Endothelial C4d deposition and the C4d score are used as reliable markers for diagnosing AMR in renal transplantation but remain controversial in lung transplantation. However, there was a report that diffuse C4d deposition in endothelial capillaries was correlated with microvascular inflammation and graft dysfunction [44]. Almost all of the mice in this study had C4d-positive AMR, and aCD20 treatment may have had a specific effect on C4d-positive AMR. C4d-negative cases have similar clinical presentations and outcomes as C4d-positive cases [45]. In short, the clinical significance of the presence or absence of C4d deposition in lung transplantation is currently poorly understood. The elucidation of complement activity is vital for the suppression of AMR [16,17]. C4d-negative and C4d-positive AMR may differ in pathophysiology, and the effect of aCD20 may vary depending on the presence or absence of C4d deposition.

The present study had some limitations. First, the lung transplantation model used in this study was major histocompatibility complex-mismatched transplantation, an acute rejection model. Therefore, the model did not evaluate chronic rejection, such as CLAD. Mimura et al. has reported on a histocompatibility complex mismatch combinations model to show histopathology for bronchiolitis obliterans in all allogeneic grafts [46]. Shiina et al. used skin-graft-induced pre-sensitization models to show enhancement of AMR [17]. Had we utilized these models, we would have been able to evaluate AMR and CLAD. Second, although the effects of aCD20s were confirmed by pathological findings and DSA, the data are insufficient to explain the pathogenesis. AMR is classified as clinical and subclinical, but blood gas analysis and respiratory function were not performed. The A scores in this experiment were assessed by the degree of perivascular cellular infiltration, but inflammatory cell types were not identified. In addition, the determination of the B-cell suppressive effect of aCD20 showed a decrease in the B cell population but did not identify the type of B cells or measure organ weights or cell counts. The cytokine analysis in this study was based only on serum samples. To understand the local reaction, a cytokine analysis of bronchoalveolar lavage fluid and cell culture of lung tissue was also needed. Furthermore, cytokine and DSA evaluation should be performed periodically to reveal dynamic mechanisms.

To our knowledge, this is the first report on the efficacy of single additional use of aCD20 treatment for preventing AMR after lung transplantation. CNI immunosuppression combined with aCD20 therapy effectively suppressed AMR in lung transplantation by reducing DSA and complement activation. Further basic and clinical studies are warranted to verify the efficacy of aCD20s in lung transplantation.

## Figures and Tables

**Figure 1 life-13-02042-f001:**
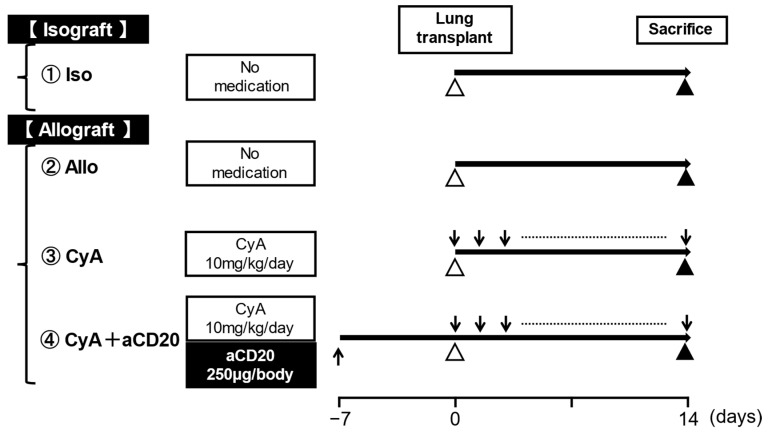
Experimental scheme. Isograft is C57BL/6 to C57BL/6, and allograft is BALB/c to C57BL/6. In the Iso and Allo control groups, mice did not receive any medications. In the CyA and CyA+aCD20 groups, CyA was administered daily subcutaneously, and aCD20 was administered intravenously 7 days before transplantation. Mice in all groups were sacrificed 14 days after lung transplantation. Each group consisted of more than five animals. The down arrow is administration of CyA, the up arrow is administration of aCD20, the white arrowhead is lung transplant, and the black arrowhead is sacrifice. aCD20, anti-CD20 antibody; CyA, cyclosporine A.

**Figure 2 life-13-02042-f002:**
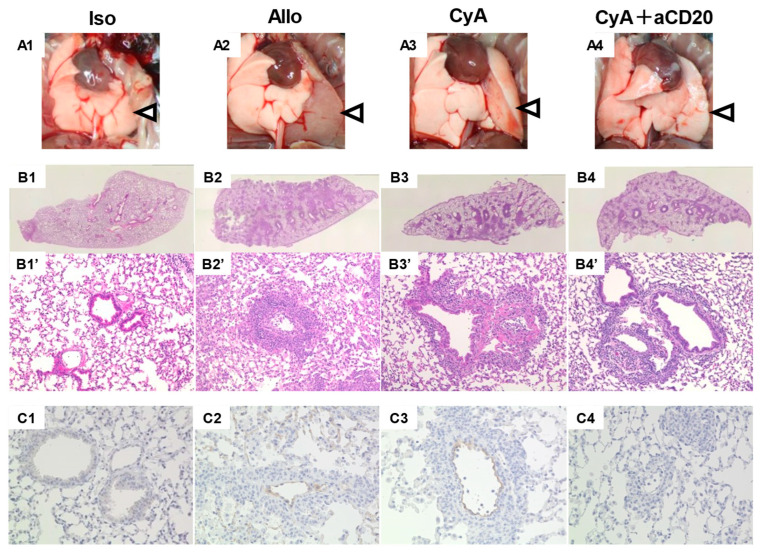
Representative macroscopic and pathological findings of lung grafts at the time of sacrifice. Upper photographs (**A1**–**A4**) show macroscopic findings (arrowheads point to the transplanted lung grafts). Middle microphotographs (**B1**–**B4** and **B1′**–**B4′**) show HE staining. Magnification, 4× and 20×. Lower microphotographs (**C1**–**C4**) show C4d immunostaining. Magnification, 40×. In the macrographic comparison, while no major macroscopic changes were detected in the Iso control group, the Allo control group showed color changes due to poor lung congestion; the CyA group showed little improvement, and the CyA+aCD20 group showed significant improvement. In the Iso control group, no cellular infiltration or tissue damage was observed. In the other groups, the lungs showed perivascular and intra-alveolar infiltration of inflammatory cells. In the Allo control and CyA groups, the lungs showed linear deposits of C4d in the capillary and vascular endothelium, but no detectable C4d deposition was observed in the CyA+aCD20 group. aCD20, anti-CD20 antibody; CyA, cyclosporine A; HE, hematoxylin and eosin.

**Figure 3 life-13-02042-f003:**
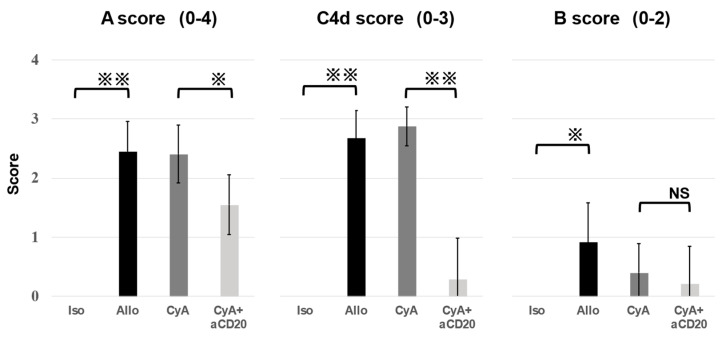
Histological grading of lung grafts. A-grade scores and C4d scores were significantly lower in the CyA+aCD20 group than in the CyA group (A score, 2.7 ± 0.46 vs. 1.8 ± 0.87, *p* = 0.0067; C4d score, 2.88 ± 0.33 vs. 0.29 ± 0.70, *p* = 0.00063). There were no differences in B-grade scores between the CyA+aCD20 and CyA groups. Data are presented as mean ± standard division (SD) (n = 6–11). ※, *p* < 0.05; ※※, *p* < 0.01; NS, no significant difference; aCD20, anti-CD20 antibody; CyA, cyclosporine A.

**Figure 4 life-13-02042-f004:**
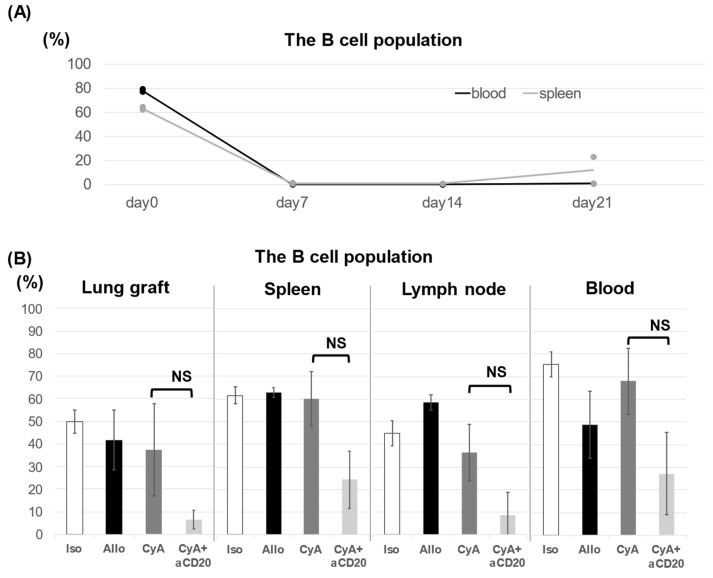
Percentage of B cells in the lung graft, spleen, lymph nodes, and peripheral blood. The B cell population, expressed as a percentage of total lymphocytes in the spleen and blood, was analyzed for 1 to 3 weeks after the administration of the aCD20 (n = 2, respectively, plot data is median) (**A**). The B cell population in the lung, spleen, lymph node, and blood was lower in the CyA+aCD20 group than in the CyA group, but not significant (graft lung, 37.49 ± 20.38 vs. 6.60 ± 4.20, *p* = 0.112; spleen, 60.11 ± 12.24 vs. 24.35 ± 12.64, *p* = 0.0509; lymph node, 8.47 ± 10.41 vs. 36.26 ± 12.45, *p* = 0.0509; blood, 27.16 ± 18.15 vs. 67.94 ± 14.51, *p* = 0.203). (**B**). Data are presented as mean % ±SD (n = 5). NS, no significant difference; aCD20, anti-CD20 antibody; CyA, cyclosporine A; SD, standard deviation.

**Figure 5 life-13-02042-f005:**
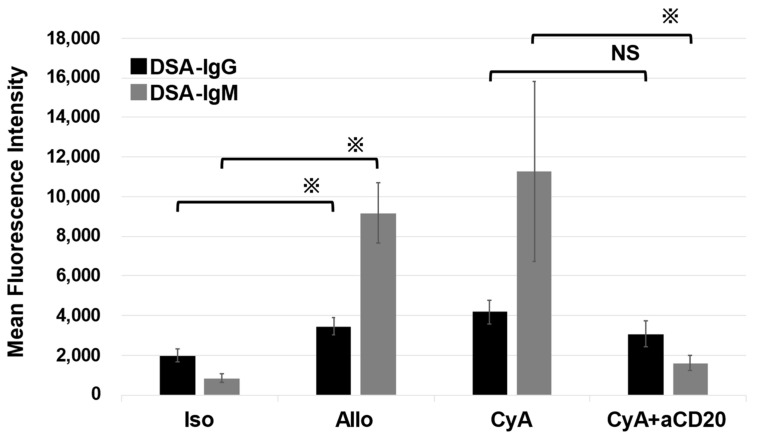
Analysis of DSA after lung transplantation in each group. The MFI of DSA-IgM was significantly lower and that of DSA-IgG showed a trend toward a decrease in the CyA+aCD20 group compared with those in the CyA group (IgM, 3071.2 ± 666.1 vs. 4176.8 ± 613.2, *p* = 0.045; IgG, 1605 ± 385.0 vs. 11283.8 ± 4534.1, *p* = 0.29). Data are presented as mean MFI ± SD (n = 5). ※, *p* < 0.05; NS, no significant difference; aCD20, anti-CD20 antibody; CyA, cyclosporine A; DSA, donor-specific antibodies; MFI, mean fluorescence intensity; SD, standard deviation.

**Figure 6 life-13-02042-f006:**
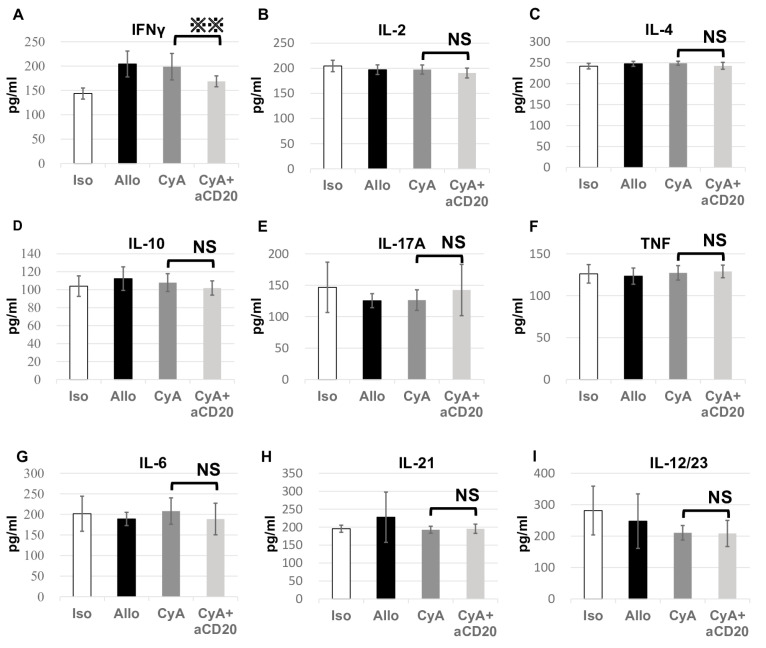
Cytokine levels in serum for each experimental group. Serum cytokine levels were measured in triplicate 14 days after transplantation. IFN-γ levels increased in the Allo control and CyA groups compared with those in the Iso control group, whereas the levels significantly decreased in the CyA+aCD20 group compared with those in the CyA group (**A**). There were no significant differences in the other cytokines among the groups (**B**–**I**). Data are presented as mean±SD (n = 5, triplicate analyses were performed to eliminate variation in the values). ※※, *p* < 0.01; NS, no significant difference; aCD20, anti-CD20 antibody; CyA, cyclosporine A; IFN, interferon; SD, standard deviation.

**Table 1 life-13-02042-t001:** a. Comparison of scores for the Allo and CyA groups. b. Comparison of scores for the CyA and CyA+aCD20 groups.

(a)			
	**Allo**	**CyA**	***p*-Value**
A score	2.45 ± 0.50	2.40 ± 0.49	0.99
C4d score	2.67 ± 0.47	2.88 ± 0.33	0.76
B score	0.91 ± 0.67	0.40 ± 0.49	0.31
**(b)**			
	**CyA**	**CyA+aCD20**	***p*-Value**
A score	2.40 ± 0.49	1.55 ± 0.50	0.030
C4d score	2.88 ± 0.33	0.29 ± 0.70	0.0028
B score	0.40 ± 0.49	0.22 ± 0.63	0.65

aCD20, anti-CD20 antibody; CyA, cyclosporine A.

## Data Availability

The data used in this manuscript are available upon request.

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
