# Peer review of "Anti-CD20 Antibody and Calcineurin Inhibitor Combination Therapy Effectively Suppresses Antibody-Mediated Rejection in Murine Orthotopic Lung Transplantation"

_life, 2023, doi:10.3390/life13102042_

Round 1
Reviewer 1 Report
Pleas indicate number of mice/group
Fig2. Are these representative histologies of
n=?
Fig 4 Add SD as % of the B cell values
Minor
Author Response
Response to Comments
Thank you very much for taking the time to review this manuscript. Please find the detailed responses below and the re-submitted files (added parts are highlighted).
We have also made the current corrections pointed out by other reviewers.
Comments 1:
Please indicate number of mice/groups?
Fig2; Are these representative histologies of
n=?
Response 1:
Thank you for pointing this out.
We have successfully executed lung transplantation procedures and subsequently prepared paraffin sections. These procedures involved 6 cases of isogenic (Iso), 11 allogenic (Allo), 10 cyclosporine A (CyA), and 9 cyclosporine A combined with anti-CD20 (CyA+aCD20). Please note that the sample sizes (n=6-11) are duly recorded within the manuscript. Figures 3, 4, 5, and 6 exhibit varying numerical values for each experimental group, and the relevant numerical data (n=) are meticulously provided in the respective figure legends. It is imperative to acknowledge that the images presented in Figure 2 are indeed representative of the histological images encompassing these groups. In response to another reviewer's feedback concerning magnification, we have ensured that the original images were utilized, with uniform magnification applied to each category.
Change:
We have emphasized the quantification of mice numbers (Lines 223, 247, 262, and 276) and have subsequently revised Figure 2 in the revised manuscript.
Comments 2:
Fig 4 Add SD as % of the B cell values.
Response 2:
We extend our gratitude for your observation. In Figure 4(A), where n=2, we have depicted data points for each day based on this sample size.
Change:
We have made the necessary revisions to Figure 4(A) within the manuscript.
Reviewer 2 Report
This study aimed to investigate the effectiveness of using anti-CD20 antibodies (aCD20s) alongside calcineurin inhibitors in preventing antibody-mediated rejection (AMR) in a murine orthotopic lung transplantation model. The finding is valuable. There are minor comments to improve the manuscript. The detailed remarks are suggested below.
Iine72 “2. Objective” does not necessary.
Line 90. What constitutes a dose of Buprenorphine?
What was the age of the donors?
What was the gender of the animals in this study?
Please provide a scale bar in fig 2
Please improve a magnification of fig2
Fig2 should be labeled as A1, A2…etc
The error bar for Figure 4 is quite large. Please provide a discussion regarding it.
Author Response
Response to Comments
Thank you very much for taking the time to review this manuscript. Please find the detailed responses below and the re-submitted files(added parts are highlighted). We have also made the current corrections pointed out by other reviewers.
Comments 1:
Iine72 “2. Objective” is not necessary.
Response 1:
The inclusion of "2. Objective" has been deemed unnecessary, and accordingly, we have updated the manuscript.
Comments 2:
Line 90. What constitutes a dose of Buprenorphine?
Response 2:
We appreciate your inquiry.
Change:
The buprenorphine dosage and surgical technique particulars have been incorporated into the manuscript.
"Buprenorphine (0.1 mg/kg) and saline solution (1 ml) were subcutaneously administered for pain management and rehydration during general anesthesia." (Lines 89-91, page 2).
Comments 3:
What was the age of the donors?
What was the gender of the animals in this study?
Response 3:
We acknowledge your queries. The donor mice were in the age range of 6–12 weeks and were of the male gender.
Change:
We have included the following sentences.
"All experiments utilized male mice."
" The BALB/c(H2d) and C57BL/6 (H2b) mice were utilized as donors, aged between 6–12 weeks (body weight, 24–32 g)." (Lines 77-82, page 2)
Comments 4:
Please provide a scale bar in fig 2
Please improve a magnification of fig2
Fig2 should be labeled as A1, A2…etc
Response 4:
We appreciate your feedback. We concur with your suggestions. Regrettably, we encountered some challenges in addressing certain issues. As it proved too intricate to incorporate a scale bar into the figure, we endeavored to standardize the image scale and provide magnification details. The variation in scale for Figure 2 in the first manuscript arose because it was generated by extracting characteristic regions from the original image.
The composite image (B1-4) exhibits slight scale differences as it was formed by amalgamating magnified 4x images from various sources.
We changed the images because we have given the unavailability of original data for certain images.
Change:
We have changed the Figure 2. (page5)
By utilizing the original image for each component and ensuring appropriate sizing and magnification notations, we have labeled the figure numbers.
Comments 5:
The error bar for Figure 4 is quite large. Please provide a discussion regarding it.
Response 5:
Thank you for pointing this out. We added the sentences in Discussion.
Change:
In Figure 4A, at day 21, a notable disparity in the spleen's B-cell population was observed, with percentages of 0.96% and 23%. Referring to the original paper [25], it is conceivable that the extended effect of the anti-CD20 antibody may have contributed to this variation. Factors such as residual drugs within the syringe or needle or individual variations among mice may have played a role in this outcome. (Lines 313-320, page 10).
The differential B cell suppression seen in Figure 4B for the CyA+aCD20 group compared to Figure 4A may be attributed to discrepancies in drug administration or the impact of surgical intervention. Furthermore, Figure 4B illustrates larger error bars in the case of three major alloantigen strain mismatch transplant models compared to the Iso group. Some of these error bars in Figure 4B exhibit substantial magnitude, irrespective of drug administration, implying that major alloantigen strain mismatch transplant procedures likely influence the B-cell percentage beyond the effect of B cell suppression by aCD20. (Lines 323-329, page 10).
Reviewer 3 Report
In this review, the authors tested the efficacy of anti-CD20 treatment in a mouse model of antibody-mediated rejection (AMR) following lung transplant. Immunosuppression with calcineurin inhibitors is important following lung transplantation, but it is insufficient to prevent or treat AMR. Anti-CD20 treatment has improved clinical outcomes in other forms of transplantation (such as renal transplantation), but it’s efficacy for lung transplantation is not well understood.
In this study, a MHC-mismatched transplantation model was used. Four experimental groups were included: an isograft control, an allogeneic transplant control with no medication, an allogeneic transplant group treated with cyclosporine A (a calcineurin inhibitor), and and an allogeneic transplant group treated with cyclosporine A + murine anti-CD20. The authors report histological findings, B-cell levels, levels of donor-specific IgG and IgM, as well as cytokine levels for all four experimental groups. The combined cyclosporin A + anti-CD20 treatment group had reduced donor specific IgM, B-cell levels, IFN-γ levels, and complement activation compared to the other experimental groups. The macroscopic appearance of the transplanted lung tissue also looked better in this group compared to the other two allogeneic transplantation groups. These results indicate that anti-CD20 treatment might have beneficial effects for preventing AMR after lung transplantation.
I have the following questions and comments for the authors:
(1.) In section 3.7 (lines 169 to 172, page 4), the authors mentioned that values that deviated by more than 20% of the mean value were excluded, and the mean value was recalculated. What is the basis for excluding data points that deviated by more than 20%?
(2.) In the statistical analysis section, the authors mention that they used the Mann-Whitney U test or the t-test for pairwise comparisons (lines 174 to 175, page 4). However, these methods are appropriate to use only when there are only two experimental groups. It is NOT appropriate to use these methods for statistical analysis in this study because there are four experimental groups and not two. Please reanalyze the quantitative data (reported in Figures 3, 4, 5, 6) using ANOVA with an appropriate post-hoc test for multiple comparison(such as Tukey’s HSD or Dunnett’s), and report the updated results + interpretation.
(3.) Please include a table that summarizes the data included in lines 203 to 216 (pages 5 and 6) instead of including it in line with the text. This can make it easier for readers to absorb the information being presented.
(4.) Figure 4A has an N of 2, but the authors report only the median. Since there are only 2 data points for each time point, please plot both of them instead of reporting just the median. It will be valuable to see the variability between the two data points that were collected at each time point.
(5.) The caption for Figure 6 (page 9) mentions that n = 5, and triplicate analysis was used to eliminate variation in the values. Please clarify how the triplicates were used in the statistical analysis. Did you use n = 15? Or did you take the mean value of the triplicates for each animal, and then use n = 5?
(6.) Figure 4A (n = 2) shows that the B cell population in the blood and spleen went down very significantly in the first week and stayed down for two weeks following that. However, in Figure 4B, the reduction in the blood and spleen of the CyA + aCD20 group (n = 5) is much more modest compared to Figure 4A. Why did the 5 animals in Figure 4B have such different results compared to the two animals in Figure 4A?
(7.) The authors explain several limitations of the study in lines 359 to 368 of page 11. If the authors were aware of these limitations, then why were these not addressed while the study was being carried out (for example, why did you not do a blood gas analysis, measure respiratory function, or identify the inflammatory cell types in perivascular infiltrates)?
No comments.
Author Response
Response to Comments
Thank you very much for taking the time to review this manuscript. Please find the detailed responses below and the re-submitted files (added parts are highlighted). We have also made the current corrections pointed out by other reviewers.
Comment 1:
In section 3.7 (lines 169 to 172, page 4), the authors mentioned that values that deviated by more than 20% of the mean value were excluded, and the mean value was recalculated. What is the basis for excluding data points that deviated by more than 20%?
Response 1:
Thank you for pointing this out. As you said, there was no evidence, but one example of how to consider outliers, which was easy to calculate. However, in this experiment, numbers were small, and We thought that calculations using standard deviations and quartiles were not appropriate. As a result, there were no outliers.
Comment 2:
In the statistical analysis section, the authors mention that they used the Mann-Whitney U test or the t-test for pairwise comparisons (lines 174 to 175, page 4). However, these methods are appropriate to use only when there are only two experimental groups. It is not appropriate to use these methods for statistical analysis in this study because there are four experimental groups and not two. Please reanalyze the quantitative data (reported in Figures 3, 4, 5, 6) using ANOVA with an appropriate post-hoc test for multiple comparison (such as Tukey’s HSD or Dunnett’s) and report the updated results + interpretation.
Response 2:
Thank you for pointing this out. Indeed, the presence of four experimental groups is noted. However, it was of particular interest to conduct a statistical analysis comparing solely the CyA and CyA+CD20 groups, as these two groups represent a critical point of comparison. The Iso group, which demonstrated no rejection, served as a validation of the surgical technique's adequacy, while the Allo group, devoid of drug intervention, exhibited rejection compared to the Iso group. We believed it prudent to abstain from extensive analysis encompassing all four groups, recognizing that the situation at hand warranted a more focused examination, distinct from assessing the relationship among all four groups.
Comment 3:
Please include a table that summarizes the data included in lines 203 to 216 (pages 5 and 6) instead of including it in line with the text. This can make it easier for readers to absorb the information being presented.
Response 3:
Thank you for pointing this out. We agree with you.
Change:
We have created the new tables. (Table 1a and 1b, page6)
Comment 4:
Figure 4A has an N of 2, but the authors report only the median. Since there are only 2 data points for each time point, please plot both of them instead of reporting just the median. It will be valuable to see the variability between the two data points that were collected at each time point.
Response 4:
Thank you for pointing this out. We agree with you.
Change:
We altered the Figure 4A and added the sentences in Result and Discussion.
We plotted two points on each day in Figure 4A. (Figure 4, page 7)
“The B cell populations were 0.96% and 23% in the spleen at 3 weeks after aCD20 administration, a significant difference between the two cases.” (Lines 234-235, page 7)
Comment 5:
The caption for Figure 6 (page 9) mentions that n = 5, and triplicate analysis was used to eliminate variation in the values. Please clarify how the triplicates were used in the statistical analysis. Did you use n = 15? Or did you take the mean value of the triplicates for each animal, and then use n = 5?
Response 5:
We used n = 15.
Given that three samples were drawn from each individual mouse, inter-sample correlation exists within the values. Nonetheless, we determined that this approach did not compromise the integrity of the statistical analysis.
Comment 6:
Figure 4A (n = 2) shows that the B cell population in the blood and spleen went down very significantly in the first week and stayed down for two weeks following that. However, in Figure 4B, the reduction in the blood and spleen of the CyA + aCD20 group (n = 5) is much more modest compared to Figure 4A. Why did the 5 animals in Figure 4B have such different results compared to the two animals in Figure 4A?
Response 6:
Thank you for pointing this out.
We believe that the most significant reason is that the impact of the surgical invasion especially major alloantigen strain mismatch transplant outweighs the impact of the B-cell suppression by aCD20.
Change:
To address this disparity, we have incorporated the following statements into the Discussion.
"The observed variation in Figure 4B, where the CyA+aCD20 group exhibited milder B cell suppression compared to Figure 4A, may be attributed to factors such as potential errors in drug administration or the influence of surgical intervention. Furthermore, Figure 4B highlights larger error bars in the context of three major alloantigen strain mismatch transplant models, in contrast to the Iso group. It is evident that in Figure 4B, certain error bars exhibit substantial magnitude irrespective of drug administration, suggesting that major alloantigen strain mismatch transplant procedures are likely to impact the percentage of B cells beyond the effects of B cell suppression by aCD20." (Lines 323-329, page 10)
Comment 7:
The authors explain several limitations of the study in lines 359 to 368 of page 11. If the authors were aware of these limitations, then why were these not addressed while the study was being carried out (for example, why did you not do a blood gas analysis, measure respiratory function, or identify the inflammatory cell types in perivascular infiltrates)?
Response 7:
We offer our sincere apologies for the absence of certain data and the limitations outlined in the manuscript. The limitations were acknowledged, but practical constraints impeded their resolution during the study. Regrettably, we lacked the requisite equipment for tasks such as blood gas analysis and respiratory function measurement, and conducting these tests was only feasible on the day of the experiment. These limitations came to light post-experiment, as their importance in evaluating lung transplantation results became apparent. Our inability to conduct additional experiments was primarily due to the need to sacrifice mice, as well as constraints in time and resources. We commit to incorporating these aspects in future studies should the opportunity arise with added medications or experiments.
Round 2
Reviewer 3 Report
Thank you for providing detailed responses to my previous comments. I appreciate most of the responses and the changes that were made to the manuscript. However, I disagree with the reasoning provided for not conducting an ANOVA followed by a post-hoc test in Response 2.
The authors’ response is that the particular interest of the statistical analysis is to solely compare the CyA group with the CyA+CD20 group, which is why the t-test and the Mann-Whitney U test were used instead of an ANOVA. If this is indeed the main focus of the statistical analysis, then this should be made clear in the main text, and Figures 3, 4, 5, and 6 should only show the results for these two groups instead of showing the results for all four groups.
These figures currently show all four groups, and they also show the statistical comparisons for the Iso and Allo groups (and not just the comparison for the CyA group with the CyA+CD20 group). If the authors keep the current figure format, then it unintentionally misrepresents the statistical analysis and gives the reader the impression that all four groups were correctly compared (which would be to use the ANOVA).
Therefore, my recommendation to the authors would be to either (a) Clarify that the CyA group and the CyA+CD20 group are the only two groups of interest for the statistical analysis, use the t-test/Mann-Whitney U test, and show only the comparison of these two groups in Figures 3 to 6, or (b) Show all four groups in Figures 3 to 6, but redo the statistical analysis using ANOVA + an appropriate post-hoc test (such as the Tukey’s test).
In this reviewer’s opinion, this manuscript is not publishable in it’s current form unless the authors address these issues with the analysis and representation of the data.
Author Response
Response to Comments
Thank you very much for taking the time to review this manuscript. Please find the detailed responses below and the re-submitted files(added parts are highlighted blue).
Comments and Suggestions:
Thank you for providing detailed responses to my previous comments. I appreciate most of the responses and the changes that were made to the manuscript. However, I disagree with the reasoning provided for not conducting an ANOVA followed by a post-hoc test in Response 2.
The authors’ response is that the particular interest of the statistical analysis is to solely compare the CyA group with the CyA+CD20 group, which is why the t-test and the Mann-Whitney U test were used instead of an ANOVA. If this is indeed the main focus of the statistical analysis, then this should be made clear in the main text, and Figures 3, 4, 5, and 6 should only show the results for these two groups instead of showing the results for all four groups.
These figures currently show all four groups, and they also show the statistical comparisons for the Iso and Allo groups (and not just the comparison for the CyA group with the CyA+CD20 group). If the authors keep the current figure format, then it unintentionally misrepresents the statistical analysis and gives the reader the impression that all four groups were correctly compared (which would be to use the ANOVA).
Therefore, my recommendation to the authors would be to either (a) Clarify that the CyA group and the CyA+CD20 group are the only two groups of interest for the statistical analysis, use the t-test/Mann-Whitney U test, and show only the comparison of these two groups in Figures 3 to 6, or (b) Show all four groups in Figures 3 to 6, but redo the statistical analysis using ANOVA + an appropriate post-hoc test (such as the Tukey’s test).
In this reviewer’s opinion, this manuscript is not publishable in it’s current form unless the authors address these issues with the analysis and representation of the data.
Response:
We appreciate your discerning observation. Since you brought this matter to our attention, we have diligently scrutinized the statistical aspects of our study. We concur with your astute insights. We have now come to the realization that employing the Mann–Whitney U test for the analysis presented in Figures 3-6 was not congruent with the precision required. We extend our profound gratitude for your gracious recommendations.
The dataset within this research exhibits modest sample dimensions, thereby leading us to regard the Kruskal-Wallis method as the most judicious choice for conducting multiple comparison tests. Furthermore, we have employed the Steel-Dwass post hoc test for pairwise comparisons between the two groups.
Change:
We have revised the sentence in 3.8 Statistical analysis as follows.
“When comparing four groups, data were evaluated using the Kruskal-Wallis test and post hoc Steel-Dwass test.” (Line 175-176, page 4)
Figures 3-6 (and figure legends) and Table 1 were confirmed and modified due to the change in P value.
Although not related to the comment, the wording was corrected. (Line 334, page 10)

Round 3
Reviewer 3 Report
Thank you for addressing the concerns expressed in the previous round of review, and for updating the statistics. I have no further questions or concerns.